# Factors Associated with Prolonged Hospital Length of Stay in Adults with Imported Falciparum Malaria—An Observational Study from a Tertiary Care University Hospital in Berlin, Germany

**DOI:** 10.3390/microorganisms9091941

**Published:** 2021-09-12

**Authors:** Bodo Hoffmeister

**Affiliations:** Department of Respiratory Medicine, Clinic-Group Ernst von Bergmann, Potsdam and Bad Belzig, 14806 Bad Belzig, Germany; bodohoff@gmx.de; Tel.: +49-33841-93297

**Keywords:** imported falciparum malaria, severe malaria, hospital length of stay, public health, epidemiology

## Abstract

Outcome of falciparum malaria is largely influenced by the standard of care provided, which in turn depends on the available medical resources. Worldwide, the COVID-19 pandemic has had a major impact on the availability of these resources, even in resource-rich healthcare systems such as Germany’s. The present study aimed to determine the under-explored factors associated with hospital length of stay (LOS) in imported falciparum malaria to identify potential targets for improving management. This retrospective observational study used multivariate Cox proportional hazard regression with time to discharge as an endpoint for adults hospitalized between 2001 and 2015 with imported falciparum malaria in the Charité University Hospital, Berlin. The median LOS of the 535 cases enrolled was 3 days (inter-quartile range, IQR, 3–4 days). The likelihood of being discharged by day 3 strongly decreased with severe malaria (hazard ratio, HR, 0.274; 95% Confidence interval, 95%CI: 0.190–0.396) and by 40% with each additional presenting complication (HR, 0.595; 95%CI: 0.510–0.694). The 55 (10.3%) severe cases required a median LOS of 7 days (IQR, 5–12 days). In multivariate analysis, occurrence of shock (adjusted HR, aHR, 0.438; 95%CI 0.220–0.873), acute pulmonary oedema or acute respiratory distress syndrome (aHR, 0.450; 95%CI: 0.223–0.874), and the need for renal replacement therapy (aHR, 0.170; 95%CI: 0.063–0.461) were independently associated with LOS. All patients survived to discharge. This study illustrates that favourable outcomes can be achieved with high-standard care in imported falciparum malaria. Early recognition of disease severity together with targeted supportive care can lead to avoidance of manifest organ failure, thereby potentially decreasing LOS and alleviating pressure on bed capacities.

## 1. Introduction

Falciparum malaria has a unique pathophysiology with exponential growth of parasite biomass every 48 h, reduced deformability and sequestration of infected (IE) as well as non-infected erythrocytes in the microcirculation, and systemic inflammation and endothelial dysfunction [1,2]. This unique pathophysiology can lead to various life-threatening complications, the criteria for severe malaria (Table 1). These complications may develop suddenly, even after the initiation of an effective therapy [3]. Although in countries where the disease is endemic widespread use of intravenous artesunate has demonstrated a strong impact on survival [4], mortality from severe falciparum malaria is still substantial [4,5,6,7]. Depending on transmission intensity and patient age, it can exceed 35% in patients >~50 years of age, even with appropriate treatment [4,5]. Successful management is challenging in settings with limited resources [6,7].

In Europe, falciparum malaria occurs as a rather rare imported disease. In contrast to regions where the disease is endemic, the affected patient population is more heterogeneous, consisting of malaria-naïve travellers, immigrants with waning semi-immunity visiting friends and relatives (VFR-immigrants) in their home countries, and travellers residing in regions of endemicity. As severe and fatal courses are more likely to occur among these patients, hospitalization is generally recommended. In the European countries mostly affected, namely, France, the United Kingdom, and Germany, outcome of imported falciparum malaria is considerably better compared to many high- and low-transmission areas, which is primarily the result of a higher standard of care. When treatment takes place in specialized centers, mortality rates less than 5% can be achieved [8,9,10].

Standard of care depends on the entirety of the available medical resources, such as pharmaceutics, diagnostic interventions, hospital and intensive care unit (ICU) beds, technical organ support, and, in particular, trained critical care staff including respiratory therapists, dialysis specialists, and ICU nurses [11]. Throughout the world, the COVID-19 pandemic continues to have a dramatic impact on the availability of these resources [12]. As a result, resource allocation, and in particular the availability of sufficient ICU bed capacities, has become an issue of critical importance, even in countries with resource-rich healthcare systems such as Germany [13]. Although the world-wide non-essential travel ban during the pandemic has led to a substantial decrease in case numbers of vector-borne diseases imported to European countries [14], these numbers are likely to increase as soon as international travel re-intensifies. So far, little is known about the conditions that consume most resources when managing patients with imported falciparum malaria [10]. Identifying factors associated with hospital length of stay (LOS), which is a core indicator for outcome studies and quality-of-care analyses, can give valuable insights. Strategies improving management will likely result in both improved outcomes and decreased LOS, thereby alleviating pressure on bed capacities.

**Table 1 microorganisms-09-01941-t001:** Severe imported falciparum malaria was defined according to the 2014 World Health Organization definition with minor modifications.

Criterion	Definition
Hyperparasitaemia	>10% parasitized erythrocytes ^1^
Jaundice	Plasma or serum bilirubin >3 mg/dL with parasitaemia > 100,000/µL
Acute pulmonary oedema (APO)	Radiologically confirmed and/or oxygen saturation on room air < 92% with respiratory rate >30/min.
Acute respiratory distress syndrome (ARDS)	Lung injury within 1 week of admission with progression of respiratory symptoms; bilateral opacities on chest imaging not explained by other lung pathologies; respiratory failure not explained by heart failure or volume overload; PaO_2_/FiO_2_ ≤ 300 mmHg under a minimum PEEP of 5 cmH_2_O (applied by non-invasive or invasive ventilation)
Decompensated shock	Systolic blood pressure < 80 mmHg with need for norepinephrine dosages >0.05 µg/kg/min. to maintain mean arterial blood pressure > 65 mmHg despite adequate hydration
Significant bleeding	Including recurrent or prolonged bleeding from the nose, gums, venepuncture sites, haematemesis, or malaena
Coma	Glasgow coma scale (GCS) < 11
Renal impairment	Plasma or serum creatinine >3 mg/dL or blood urea > 120 mg/dL
Metabolic acidosis	Base deficit >8 mmol/L and/or bicarbonate <15 mmol/L and/or venous plasma lactate ≥5 mmol/L or ≥45 mg/dL
Severe malarial anaemia	Haemoglobin level < 7 g/dL and/or haematocrit < 20% with parasitaemia > 0.5%
Hypoglycaemia	Blood glucose level < 40 mg/dL
Convulsions	>2 convulsions within 24 h

^1^ In contrast to a previous publication [13], a threshold of >10% instead of >5% for defining hyperparasitaemia was used.

The aim of the present study was to identify factors associated with prolonged LOS among patients with imported falciparum malaria treated in a specialized center in Berlin, Germany, in the pre-COVID-19 era.

## 2. Materials and Methods

### 2.1. Data Collection

All adult patients (≥18 years) hospitalized with imported falciparum malaria between 1 January 2001 and 31 December 2015 in the Department of Infectious Diseases and Pulmonolgy of the Charité, Berlin, a tertiary care university hospital, were enrolled (Figure 1). During the study period, 17 patients were treated more than once at the Charité University Hospital, Berlin. In order to avoid repeated analysis of the same individual, only the first falciparum malaria episodes of these patients were included. Cases with incomplete records were also excluded. Data on demographics, travel history, full medical history, including prior malaria episodes, current medication, and results of physical examination, and laboratory and radiologic investigations were retrieved from standardized electronic files, which were available for all patients. Seriousness of underlying co-morbidity was retrospectively weighed by calculating an age-adjusted Charleson co-morbidity index (CA-CCI) for each patient [15]. The study represents a secondary analysis of a previous investigation [16].

### 2.2. Clinical Management

Diagnosis of falciparum malaria relied on thin and thick blood smears. Parasitaemia was expressed as percentage of parasitized erythrocytes (1% corresponding to approximately 50,000 parasites/µL). Severe malaria was defined according to the World Health Organization (WHO) 2014 definition [3] with minor modifications and under the inclusion of the actual definition of the acute respiratory distress syndrome (ARDS; Table 1) [17]. Treatment of uncomplicated malaria initially relied on atovaquone/proguanil, mefloquine, or quinine, the latter either in combination with doxycycline or clindamycin (for details of antimalarial treatment refer to Table A1, Appendix B). Artemisinin combination therapies (ACTs) became available for the Charité University Hospital in August 2002 and quickly became the standard treatment for uncomplicated disease. Therapy of severe malaria relied on quinine in combination with either doxycycline or clindamycin before artesunate became available in Germany in 2007. Since not being manufactured in accordance with European Good Manufacturing Practice and hence lacking market authorization, artesunate was only used in patients with parasitaemias ≥10% or with contraindications for quinine. Antimalarial therapy was instituted as soon as possible after establishment of diagnosis. Supportive management consisted of restrictive fluid management, vasopressor use in shocked patients, renal replacement therapy (RRT, veno-venous hemofiltration only), and non-invasive or invasive mechanical ventilation where indicated, as outlined elsewhere [18].

### 2.3. Outcome Variable

The study’s only endpoint (i.e., the dependent variable) was LOS, which was calculated in days by subtracting the day of admission from the day of discharge. Selection of covariates (i.e., the independent variables) was based on a literature search using the Cochrane Library, PubMed, and Google Scholar databases that had recently identified factors associated with malaria LOS [6].

### 2.4. Statistical Analysis

Only non-parametric tests were used. For baseline characteristics, categorical data were compared by chi^2^ test, while the Mann–Whitney U-test was used for continuous data. Length of hospital stay was markedly right-skewed (Figure 2B). As none of the patients had died or had been transferred to another hospital, there were no competing risks in the study population. Therefore, a conventional Cox proportional hazard regression with discharge after the median of LOS (i.e., by day 3) as an endpoint was considered the appropriate statistical approach [19]. In univariate analysis, associations linking each covariate to the instantaneous hazard ratio (HR) for discharge were assessed by Cox proportional risk models with censoring of all patients discharged after 3 days. To ensure the robustness of the investigation, sensitivity analyses with censoring of all patients discharged after the third quartile of LOS (i.e., after 4 days) and after 7 days (i.e., the median LOS of patients with severe malaria) were performed [20]. Only covariates with significant associations with LOS consistent in all three analyses were included in the following multivariate analysis [21]. Covariates violating the proportional hazard assumption were excluded from the multivariate analysis. For variable selection, the best subset selection method was used [22]. The results from the final multivariate model were reported as adjusted hazard ratios (aHRs). The proportional hazard assumption was again tested for each covariate in the final multivariate model and for the global model using the Schoenfeld residual test. Influential observations were tested by dfbeta values. The statistical significance level was set at 5% for all analyses. All statistical analyses were performed using R version 4.0.3 (the R foundation for Statistical Computing). Findings were reported according to the Strengthening the Reporting of Observational Studies in Epidemiology (STROBE) Statement Cohort Studies checklist (Appendix A).

## 3. Results

### 3.1. Patient Characteristics

A total of 558 adult patients with imported falciparum malaria were treated in the Department of Infectious Diseases and Pulmonology of the Charité Universtiy Hospital, Berlin, between 2001 and 2015, corresponding to 7.1% of all 7.866 imported falciparum malaria cases notified in Germany during the study period (Figure 2). Twenty-three of these cases were excluded (Figure 1). The health status on admission of the remaining 535 patients is outlined in Table 2. The majority of these patients (*n* = 393, 73.5%) had been treated with artemisinin-based regimens. In 480 cases (89.7%), the disease was uncomplicated, while 55 (10.3%) patients suffered from severe malaria. Admission to an intensive care unit (ICU) was required in 68 (12.7%) cases. Among these, 13 (2.4%) needed vasopressors for the treatment of shock. Thirteen patients (2.4%) had radiologically confirmed acute pulmonary oedema (APO) or ARDS. In four individuals (0.8%), oxygen supplementation was sufficient. Nine patients (1.7%) required mechanical ventilation for a total of 950 ventilation hours (median: 35 h, IQR: 16–150 h, range: 1.5–384 h). Four patients were non-invasively ventilated, five invasively. Indications for invasive mechanical ventilation were moderate to severe ARDS, deep coma (Glasgow Coma Scale < 8), advanced shock, multi-organ dysfunction, and cardiac arrest. In 7 (1.3%) of the 13 patients with renal impairment, RRT was required for acute kidney injury stage 3 (AKI3). Median time to initiation of RRT was 20 h. These patients were on RRT for a total of 51 days (median: 5 days, IQR 4–9 days, range: 2–20 days).

The overall outcome was favourable: weaning from mechanical ventilation was successful in all patients, renal function normalized and RRT could de discontinued in all but one individual. This patient, however, had pre-existing chronic renal failure with a baseline creatinine >10 mg/dL and progressed to end-stage renal disease as a result of the malaria. Individuals with coma did not suffer from long-term neurologic deficits, including a patient who had to be resuscitated due to cardiac arrest in the context of severe metabolic acidosis. All patients survived to discharge.

### 3.2. Length of Hospital Stay

The total LOS of the 535 cases enrolled was 2303 in-patient days, the median LOS of the cohort being 3 days (IQR 3–4 days, range: 1–62 days; Figure 2). The 480 uncomplicated cases were hospitalized for a total of 1704 days, the median LOS also being 3 days (IQR 3–4 days, range: 1–24 days). Therapy with an ACT was associated with a shorter LOS (median of 3 versus 4 days, *p* < 0.0001). Presence of a ≥1 criterion for severe malaria was strongly associated with prolonged hospitalization (Table 3) and was the single factor with the highest impact on total LOS: with a total of 599 in-patient days, the severe cases accounted for >25% of total LOS, the median LOS of these cases being 7 days (IQR 5–12 days, range: 3–62 days). In addition, the likelihood of discharge by day 3 declined by >40% with every additional malaria-specific complication present on admission (Table 3, Figure 3A). Each malaria-specific complication was significantly associated with prolonged LOS in univariate analysis, AKI3 being the individual complication with the highest individual impact on LOS (Figure 3D). In order to avoid over-fitting, the covariate presence of the ≥1 criterion for severe malaria was not included in final multivariate analysis. The proportional hazards assumption was violated for the covariate number of complications on admission. This covariate was therefore also not included in the final multivariate analysis. The covariates acidosis, renal impairment, shock, APO/ARDS, and AKI3 were included in the final multivariable model. The latter three were independently associated with prolonged LOS in multivariate analysis (Table 4), whereby these complications had only occasionally occurred separately, but more commonly as part of multi-organ involvement together with a median of 3 other complications (Figure 4).

## 4. Discussion

The present study describes a cohort of patients with imported falciparum malaria treated under favourable conditions. A high standard of care allowed all patients to survive their disease. What were the reasons for this desirable outcome? How can it be achieved in the future in view of possibly diminishing medical resources?

Falciparum malaria can lead to a broad spectrum of life-threatening complications. The presenting syndromes of this protean illness largely vary with patient age, have different prognostic implications, and can occur separately or as part of multi-organ involvement. The sequestration of infected as well as non-infected erythrocytes is a key feature of the unique pathophysiology. Disease severity is proportional to the extent of micro-vascular obstruction [23]. Most fatalities occur within the first 48 h of admission, when parasite burden is still high [4]. Metabolic acidosis and coma directly result from the sequestration of IE in the microvasculature [24] and are the strongest predictors of a fatal outcome across all age groups [4,5,25]. However, micro-vascular obstruction usually resolves within 48 h after the initiation of an effective antimalarial therapy and the related complications begin to subside [18].

Severe complications of the kidneys and lungs are relatively common features of adult falciparum malaria [26]. In addition to sequestration, indirect damage by cytokines and cell-free haemoglobin appears to play an important role in the pathogenesis of malaria-associated AKI [27]. In the landmark South East Asian Quinine Artesunate Malaria Trial (SEAQUAMAT), the largest randomized trial on severe malaria so far, renal failure occurred in 40% of cases and was the complication with the highest individual mortality rate (38%) [4]. In a smaller, yet more recent observational study of adult severe malaria, 58% of patients had AKI, 40% of whom died, accounting for 71% of overall mortality [27]. In such fatal cases, AKI is frequently anuric and typically occurs as part of multi-organ involvement, together with coma, shock, and/or respiratory failure [28]. In survivors, complete recovery of renal function is the rule. With no less than 17 ± 6 days, however, this process requires time [28,29].

While micro-vascular obstruction can be rapidly reversed by effective antimalarials, this does not hold true for other systemic effects of the disease, such as inflammation and endothelial activation. Plasma levels of angiopoetin-2 (Ang-2), an autocrine regulator of endothelial inflammation promoting an increase in capillary permeability, correlate with disease severity. Significantly higher Ang-2 levels have been detected in patients with four severity criteria compared to individuals with only two [30]. Elevated Ang-2 plasma levels have been demonstrated for up to 4 weeks after therapy [31]. Interstitial oedemas lead to organ dysfunction, the organ system most vulnerable to the effects of increased capillary leakiness being the lungs [32]. On-going systemic inflammation and accumulation of fluids, especially in cases with simultaneous renal failure, explain why respiratory failure from APO and ARDS can occur at any time in the course of severe falciparum malaria, even after complete parasite clearance [26,33]. The SEAQUAMAT trial largely contributed to the current knowledge on the most important prognostic factors in severe falciparum malaria. High-quality imaging studies and regular monitoring of blood gas analyses, however, were not available for all patients. Respiratory distress was therefore defined on the basis of the clinical parameters respiratory rate and pulse oximetry. A number of respiratory complications due to increased capillary permeability may therefore have been missed, leading to some degree of underrepresentation of the true prognostic significance of APO and ARDS.

Mortality in patients with shock was 16% in the original SEAQUAMAT trial [4]. Shock not only contributes to interstitial oedema formation and AKI but also causes myopathy of limb and respiratory muscles. Malaria itself and its treatment can add to generalized muscle weakness [34]. Prolonged mechanical ventilation, functional impairment, and exercise limitation can be the consequences, resulting in increased mortality and prolonged lengths of ICU and hospital stays [35].

Importantly, the risk of death sharply increases with multi-organ involvement in falciparum malaria: while mortality is 9.5% in patients with a single complication, it reaches 50% in patients with >5 complications [5]. In order to better predict the mortality risk of the individual patient, various prognostic models and scores have been developed for adults in recent years. All nine models available so far use neurological dysfunction as a predictor, six use respiratory failure, and five use metabolic acidosis. Renal failure and shock are also common variables in these prognostic models [36].

The results of the present study are in line with the above observations. Similar to a recent nationwide study from Sweden, the proportion of patients with severe imported falciparum malaria was around 10% [37]. Due to the high proportion of uncomplicated cases, overall LOS was short: median LOS was 3 days, and 75% of patients could be discharged within 96 h of admission. Neither any of the well-characterized demographic risk factors for severe or fatal disease [38] nor any chronic medical conditions were found to be associated with prolonged LOS. Instead, disease severity proved to be the major determinant for LOS: severe cases required a median hospitalization of 7 days. The more complications present on admission, the longer LOS had been. Interestingly, the median LOS of the cohort’s severe cases was close to the 6 days reported from low-transmission areas [6]. In this secondary analysis of the SEAQUAMAT data, the cumulative incidence of death was 20%, with the mean time to death being 2.5 days. Most patients (31.1%) had died on the first day of admission. In contrast, all patients enrolled in the present study survived their first 48 treatment hours due to effective antimalarial therapy together with high-standard supportive care. Therefore, not acidosis and coma, but the complications usually lasting longer and/or typically occurring later in the course of the disease, namely, shock, APO, ARDS, and particularly AKI3, had determined LOS. However, these complications had only occasionally occurred separately. Instead, they occurred together with a median of three other complications as part of multi-organ involvement.

These results have two implications. To achieve optimal outcomes, reliable identification of patients at the highest risk of death that will require the most resources is mandatory. Yet, externally validated scoring systems for the assessment of disease severity are unavailable for imported malaria. Therefore, the earliest recognition of severe disease, and particularly of multi-organ involvement, appears to be crucial in patients with imported falciparum malaria.

Successful management of critically ill falciparum malaria patients has two cornerstones: effective antimalarial therapy and best supportive care. The exponential disease dynamics call for swift actions. Initiation of an effective antimalarial as soon as the diagnosis is established is the most important intervention. The artemisinins achieve the fastest parasite clearance of all antimalarials. Accordingly, their use has demonstrated a strong impact on survival in both children and adults in high- and low-transmission areas. In the present analysis, ACTs were associated with a shorter LOS in uncomplicated cases. The prognostic benefit of intravenous artesunate is greatest in patients with a high parasite burden. Yet, in non-immune travellers, a survival benefit could not be found [8,9], a fact that underscores the prognostic significance of high-standard supportive care. In the present study, all patients had survived to discharge, too, independent of the antimalarial used. However, artesunate has been shown to reduce ICU and hospital length of stay in non-immune travellers [39] and is therefore the recommended therapy for severe imported falciparum malaria, too, particularly for patients with high parasitaemias.

In addition to the rapid initiation of an effective antimalarial, early recognition of (incipient) complications that immediately results in targeted interventions is needed to achieve optimal outcomes. Without RRT, malaria-associated AKI carries a high mortality of up to 75% [29,40]. If instituted early enough, RRT considerably reduces mortality [29]. The timely initiation of RRT, therefore, must not be delayed. In the present analysis, the median time to initiation of RRT was 20 h. Optimal timing requires close monitoring of renal function. Malaria-associated AKI, however, is often non-oliguric, making urine volume output an unreliable parameter [40]. Creatinine is still the best laboratory predictor of RRT requirement available and should therefore be measured on a frequent basis [40]. Avoidance of potentially nephrotoxic drugs and over-hydration can add to reducing the risk of AKI. Although most patients with severe malaria are hypovolaemic on presentation, malaria-associated AKI is not necessarily related to hypovolaemia. In fact, no evidence has been found that restrictive fluid management with crystalloid administration of 2–3 mL/kg/h worsens kidney function or tissue perfusion [7,18]. Notably, this includes shocked patients. This approach may not only reduce the formation of interstitial oedemas in the kidneys, but also help to reduce the risk of respiratory failure [32,41]. Of note, the Fluid and Catheter Treatment Trial reported an almost significant decrease in RRT requirement in the group of patients treated with conservative fluid management for ARDS [41]. In the face of the often markedly increased capillary permeability, restrictive fluid management must be considered a key intervention of supportive care. Monitoring pulse oximetry and blood gases in regular intervals and performing appropriate imaging studies with a low threshold allow for the early identification of patients at risk of respiratory failure. This enables clinicians to prompt appropriate countermeasures at the earliest possible stage such as oxygen supplementation, application of positive airway pressures, fluid restriction, diuretics, or, if indicated, mechanical ventilation. In the present study, nearly half of patients with respiratory failure had successfully been treated with non-invasive ventilation (NIV), which is an option for eligible patients with APO and mild forms of ARDS [42,43].

In summary, it appears, that hospitals caring for patients with imported falciparum malaria can calculate with roughly 10% of severe cases and that these patients will have a median LOS of about 7 days. The above outlined treatment strategies may contribute to prevent manifest organ failure and thereby possibly aid in shortening LOS.

The present study has several limitations, the main being its retrospective, single-center design and a long observation period. This limits the generalizability of the results. Yet, the study is intended to analyze an “ideal-world scenario” for the treatment of falciparum malaria in order to identify the factors that require most resources under the best possible conditions. Generalizability is, therefore, a priori limited. Importantly, the identified factors are all well recognized prognostic factors of falciparum malaria. In addition, with a caseload of >500, the analysis covered at least 7% of the total cases notified in Germany during the study period. The availability of standardized electronic files for all patients ensured a high level of data capture. Discharge after 3 days, i.e., the median LOS of the cohort, was arbitrarily chosen to define the primary outcome parameter. However, performing sensitivity analyses using other thresholds ensured the robustness of this approach. Most importantly, data on factors influencing LOS in imported falciparum malaria are scarce, and resources for critically ill patients may continue to be limited. As standardized prognostic scoring systems for the assessment of disease severity are currently unavailable, more data supporting resource allocation are needed for the best management of this complex illness.

## 5. Conclusions

Falciparum malaria is a protean illness with rapid disease dynamics. Micro-vascular obstruction plays a central role in its unique pathophysiology. Immediate initiation of an effective antimalarial is therefore of paramount importance for successful management. However, rapid parasite clearance is not the only significant task to be achieved. Adverse effects from more indirect disease mechanisms such as generalized endothelial dysfunction, systemic inflammation with increased capillary permeability, renal tubular damage, or muscle weakness may require considerably more time to resolve. The results of the present study illustrate that with a high standard of care, mortality from the disease can be kept to a minimum. Yet, due to the COVID-19 pandemic, the medical resources required to provide a high standard of care have become more scarce in many settings. Optimal supportive care can achieve two important tasks simultaneously: improve outcomes and save valuable resources. Correct assessment of disease severity at the earliest possible stage, use of a restrictive fluid management strategy, timely initiation of RRT if indicated, and avoidance of invasive mechanical ventilation if non-invasive ventilation is appropriate are key measures that will likely reduce the LOS of patients with imported falciparum malaria and thereby relieve pressure on the need for hospital and, in particular, ICU beds.

## Figures and Tables

**Figure 1 microorganisms-09-01941-f001:**
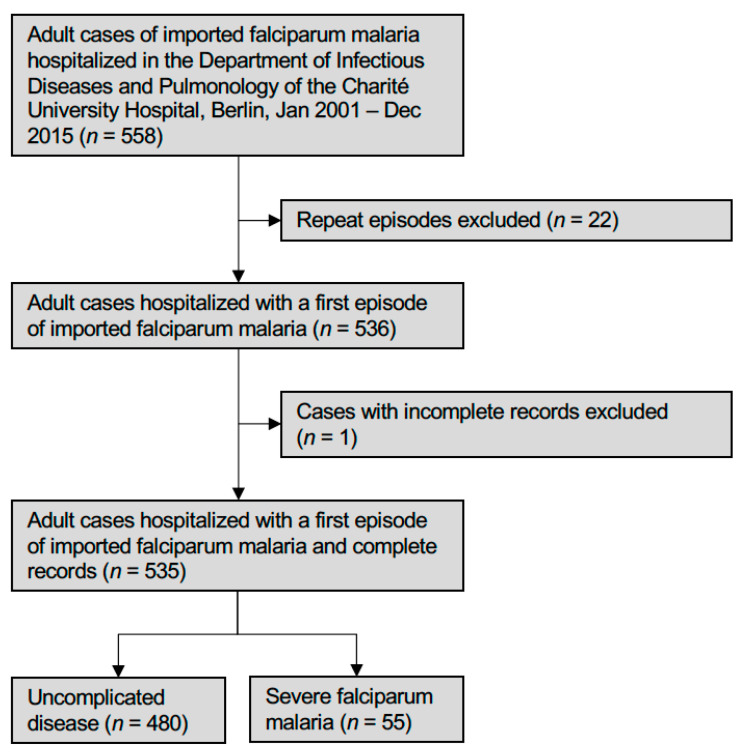
Flowchart of patients included and excluded.

**Figure 2 microorganisms-09-01941-f002:**
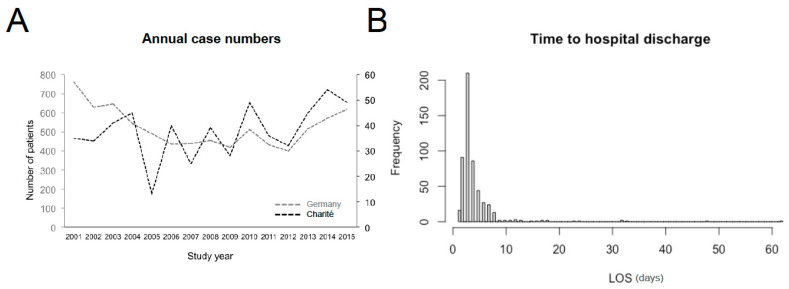
The 535 cases included in the study represented 7.1% of the total 7866 imported falciparum malaria cases notified in Germany during the study period (**A**). Note the different scales for total German cases (light grey) and cases treated in the study site (black). (**B**) Length of hospital stay in the study population.

**Figure 3 microorganisms-09-01941-f003:**
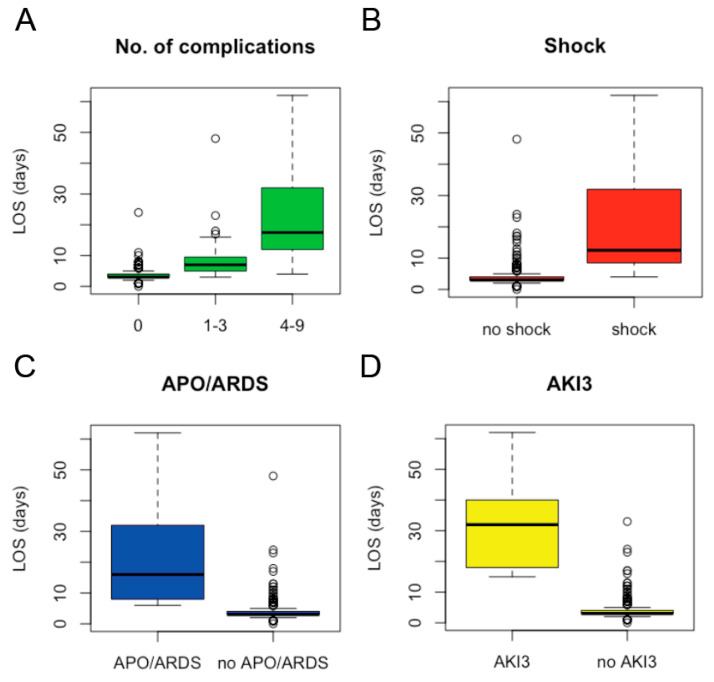
Individual factors with major influence on hospital length of stay among the 535 cases with imported falciparum malaria were the number of presenting syndromes (**A**), shock (**B**), APO or ARDS (**C**), and acute renal failure requiring renal replacement therapy (AKI3) (**D**).

**Figure 4 microorganisms-09-01941-f004:**
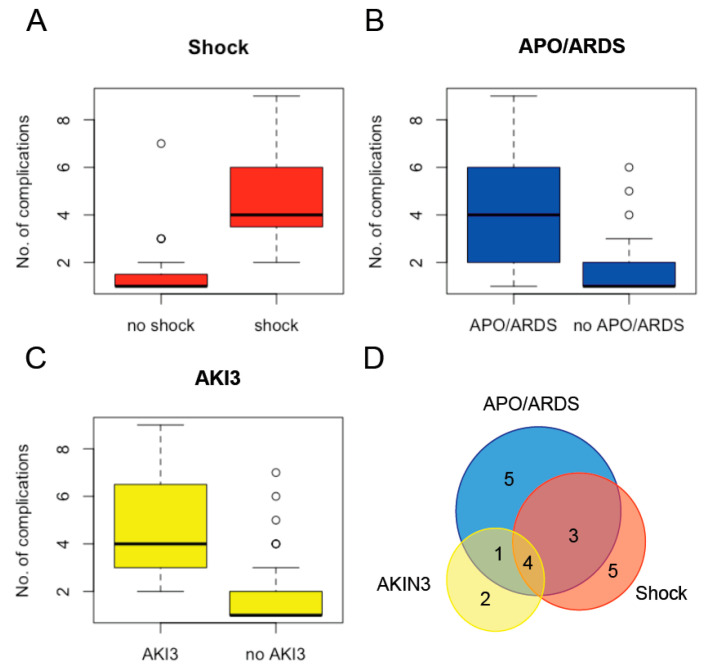
The complications associated with prolonged hospital length of stay, namely, shock (**A**), APO or ARDS (**B**), and acute renal failure requiring renal replacement therapy (AKI3) (**C**), occasionally occurred separately, but more commonly as part of multi-organ involvement together with a median of 3 other complications (**D**).

**Table 2 microorganisms-09-01941-t002:** Health status on admission of the study population.

Covariate	Whole Group (*n* = 535)	Uncomplicated Malaria (*n* = 480)	Severe Malaria (*n* = 55)	*p* Value ^1^
Demographics and history				
Age in years ^2^ (total *n* ^3^)	535	480	55	0.121
Median and IQR	37 (29;46)	37(29;45)	38 (30; 51)	
Gender ^4^ (total *n*)	535	480	55	0.018
Females, n (%)	168 (31.4)	143 (29.8)	25 (45.5)	
Males, *n* (%)	367 (68.6)	337 (70.2)	30 (54.5)	
Origin (total *n*)	535	480	55	0.409
From country, where malaria is endemic, *n* (%)	329 (61.5)	298 (62.1)	31 (56.4)	
From country, where malaria is not endemic, *n* (%)	206 (38.5)	182 (37.9)	24 (43.6)	
History of previous malaria episodes (total *n*)	535	480	55	0.158
≥1 episode, *n* (%)	163 (30.5)	149 (31.0)	12 (21.8)	
no previous episodes, *n* (%)	372 (69.5)	331 (69.0)	43 (78.2)	
Use of chemoprophylaxis (total *n*) ^5^	441	394	47	0.733
Regular use, *n* (%)	15 (3.4)	13 (3.3)	2 (4.3)	
Irregular/no use, *n* (%)	426 (96.6)	381 (96.7)	45 (95.7)	
Destination (total *n*) ^6^	523	468	55	<0.001
WHO African region, *n* (%)	507 (96.8)	459 (98.1)	48 (87.3)	
WHO Southeast Asian region, *n* (%)	16 (3.1)	9 (1.9)	7 (12.7)	
Duration from symptom onset to hospital admission, days (total *n*)				
Median and IQR	455	404	51	0.015
	4 (3;6)	4 (3;6)	5 (3;7)	
Pregnancy (total *n*)	168	143	25	0.577
Pregnant, *n* (%)	11 (6.5)	10 (7.0)	1 (4.0)	
Co-morbidities				
CA-CCI (total *n*)	535	480	55	<0.001
Median and IQR	0 (0;1)	0 (0;1)	5 (3;7)	
Hypertension (total *n*)	535	480	55	0.017
*n* (%)	43 (8.0)	34 (7.1)	9 (16.4)	
Diabetes (total *n*)	535	480	55	0.767
*n* (%)	16 (3.0)	14 (2.9)	2 (3.6)	
HIV infection (total *n*)	535	480	55	0.003
*n* (%)	15 (2.8)	10 (2.1)	5 (9.1)	
Chronic pulmonary disease (total *n*)	535	480	55	0.54
*n* (%)	13 (2.4)	11 (2.3)	2 (3.6)	
Cardiovascular disease (total *n*)	535	480	55	<0.001
*n* (%)	12 (2.2)	5 (1.0)	7 (12.7)	
Hepatitis B and/or C (total *n*)	535	480	55	0.383
*n* (%)	11 (2.1)	9 (1.9)	2 (3.6)	
Chronic renal disease (total *n*)	535	480	55	0.038
*n* (%)	10 (1.9)	7 (1.5)	3 (5.5)	
Malignancy (total *n*)	535	480	55	0.003
*n* (%)	9 (1.7)	6 (1.3)	3 (5.5)	
Obesity ^7^ (total *n*)	248	204	44	0.035
*n* (%)	24 (9.6)	16 (7.8)	8 (18.2)	
BMI in kg/m^2^ (total *n*)	248	204	44	0.793
Median and IQR	24.5 (22.4;27.2)	24.5 (22.3; 27.3)	24.4 (22.3;26.1)	
Severe malaria				
≥1 criterion (total *n*)	535	480	55	<0.001
*n* (%)	55 (10.3)	0 (0.0)	55 (100.0)	
No. of complications on admission (total *n*)	535	480	55	-
Median and IQR	0 (0;0)	0 (0;0)	1 (1;3)	
Acidosis (total *n*)	535	480	55	<0.001
*n* (%)	7 (1.3)	0 (0.0)	7 (12.7)	
Haemoglobinuria (total *n*)	535	480	55	<0.001
*n* (%)	15 (2.8)	3 (0.6)	11 (20.0)	
Renal impairment (total *n*)	535	480	55	<0.001
*n* (%)	13 (2.4)	0 (0.0)	13 (23.6)	
APO (total n)	535	480	55	<0.001
n (%)	8 (1.5)	0 (0.0)	8 (14.5)	
ARDS (total *n*)	535	480	55	<0.001
*n* (%)	5 (0.9)	0 (0.0)	5 (9.1)	
Shock (total *n*)	535	480	55	<0.001
*n* (%)	12 (2.2)	0 (0.0)	12 (21.8)	
Coma (total *n*)	535	480	55	<0.001
*n* (%)	7 (1.3)	0 (0.0)	7 (12.7)	
Seizures (total *n*)	535	480	55	-
*n* (%)	0 (0)	0 (0.0)	0 (0.0)	
Hyperparasitemia (>10%) (total *n*)	523	468	55	<0.001
*n* (%)	20 (3.8)	0 (0.0)	20 (36.4)	
Laboratory findings				
Mixed malaria ^8^ (total *n*)	523	468	55	0.127
*n* (%)	19 (3.6)	15 (3.2)	4 (7.3)	
Leucocytosis >10.5/nL on admission (total *n*)	524	471	53	<0.001
*n* (%)	12 (2.3)	4 (0.8)	8 (15.1)	
Thrombocytopenia <150/nL on admission (total *n*)	525	471	54	0.001
*n* (%)	433 (82.5)	380 (80.7)	53 (98.1)	
Thrombocytopenia <50/nL on admission (total *n*)	525	471	54	<0.001
*n* (%)	95 (17.8)	64 (13.6)	31 (57.4)	
Management				
Healthcare-associated infection ^9^ (total *n*)	535	480	55	<0.001
*n* (%)	11 (2.1)	2 (0.4)	9 (16.4)	
Mechanical ventilation (total *n*)	535	480	55	<0.001
*n* (%)	9 (1.7)	0 (0.0)	9 (16.4)	
AKI3 with need for RRT during admission (total *n*)	535	480	55	<0.001
*n* (%)	7 (1.3)	0 (0.0)	7 (12.7)	
Initial antimalarial (total *n*)	535	480	55	0.039
Artemisinin-based regimen (reference)	393 (73.5)	359 (74.8)	34 (61.8)	
Other	142 (26.6)	121 (25.2)	21 (38.2)	

^1^ Compares uncomplicated with severe cases; ^2^ Continuous variables assessed by Mann–Whitney U-test; ^3^ Indicates number of cases for whom data were available; ^4^ Categorical variables assessed by chi^2^ test; ^5^ In 94 cases (86 uncomplicated and 8 severe), information regarding use of chemoprophylaxis was not available from the medical records; ^6^ Other destinations were the WHO Region of the Americas (*n* = 5), WHO Eastern Mediterranean region (*n* = 4), WHO European region (*n* = 1), unknown (*n* = 2); ^7^ BMI > 30 kg/m^2^; ^8^ simultaneous infection with *P. falciparum* and either *P. vivax* (*n* = 7), *P. malaria* (*n* = 7), or *P. ovale* (*n* = 5); ^9^ Blood stream infections (*n* = 5), urinary tract infections (*n* = 4), and respiratory tract infections (*n* = 2); Abbreviations: AKI3, acute kidney injury stage 3; APO, acute pulmonary oedema; ARDS, acute respiratory distress syndrome; BMI, body mass index; CA-CCI, age-adjusted Charleson co-morbidity index.

**Table 3 microorganisms-09-01941-t003:** Univariate analysis of factors associated with hospital length of stay.

	Discharge after 3 Days		Discharge after 4 Days		Discharge after 7 Days	
Covariate	*p* Value	HR (95% CI)	*p* Value	HR (95% CI)	*p* Value	HR (95% CI)
Demographics and history						
Age	0.027	0.988 (0.978–0.998)	0.024	0.985 (0.971–0.998)	0.783	0.996 (0.969–1.024)
Gender	0.242	1.18 (0.894–1.557)	0.487	1.133 (0.797–1.612)	0.657	0.860 (0.442–1.673)
Origin	0.344	1.139 (0.870–1.491)	0.928	1.016 (0.718–1.438)	0.121	0.573 (0.284–1.158)
Previous malaria	0.624	1.086 (0.781–1.511)	0.249	1.278 (0.842–1.938)	0.835	0.902 (0.344–2.367)
Chemoprophylaxis	0.449	1.296 (0.662–2.540)	0.893	0.934 (0.343–2.542)	0.642	0.621 (0.084–4.614)
Destination	0.053	1.746 (0.992–3.071)	0.117	1.681 (0.878–3.216)	0.431	0.714 (0.308–1.652)
Duration from symptom onset to hospital admission	0.351	0.976 (0.927–1.027)	0.594	0.982 (0.916–1.051)	0.354	0.922 (0.776–1.095)
Pregnancy	0.861	1.083 (0.450–2.634)	0.299	1.701 (0.624–4.635)	0.354	0.922 (0.776–1.095)
Co-morbidities						
CA-CCI	0.007	0.887 (0.812–0.968)	0.022	0.885 (0.797–0.983)	0.141	0.885 (0.752–1.041)
Hypertension	<0.001	0.424 (0.263–0.684)	<0.001	0.368 (0.207–0.654)	0.058	0.450 (0.197–1.029)
Diabetes	0.629	0.830 (0.391–1.765)	0.698	1.177 (0.518–2.677)	0.774	0.746 (0.101–5.517)
HIV infection	0.186	0.674 (0.375–1.210)	0.352	0.724 (0.366–1.430)	0.83	1.123 (0.388–3.255)
Chronic pulmonary disease	0.355	1.523 (0.625–3.715)	0.165	2.272 (0.713–7.241)	1.0	-
Cardiovascular disease	0.008	0.397 (0.202–0.783)	0.028	0.442 (0.213–0.914)	0.203	0.536 (0.206–1.399)
Hepatitis B and/or C	0.059	0.453 (0.199–1.031)	0.112	0.480 (0.194–1.188)	0.465	0.636 (0.189–2.142)
Chronic renal disease	0.257	0.592 (0.239–1.467)	0.188	0.454 (0.140–1.471)	0.402	0.531 (0.120–2.338)
Malignancy	0.07	0.434 (0.176–1.068)	0.104	0.432 (0.157–1.188)	0.208	0.393 (0.092–1.681)
Obesity	0.222	0.666 (0.347–1.278)	0.910	0.962 (0.492–1.881)	0.143	2.162 (0.770–6.070)
BMI	0.794	0.994 (0.946–1.043)	0.473	1.021 (0.965–1.08)	0.069	1.086 (0.993–1.188)
Complications						
≥1 complication during admission	<0.001	0.274 (0.190–0.396)	<0.001	0.274 (0.179–0.420)	0.003	0.322 (0.151–0.688)
No. of complications on admission	<0.001	0.595 (0.510–0.694)	<0.001	0.621 (0.528–0.730)	<0.001	0.727 (0.603–0.877)
Acidosis	<0.001	0.150 (0.061–0.375)	<0.001	0.171 (0.068–0.429)	0.009	0.275 (0.105–0.724)
Haemoglobinuria	0.001	0.375 (0.208–0.675)	0.020	0.489 (0.268–0.892)	0.367	0.682 (0.297–1.568
Renal impairment	<0.001	0.230 (0.123–0.432)	<0.001	0.219 (0.114–0.420)	0.003	0.299 (0.137–0.654)
APO or ARDS	<0.001	0.234 (0.127–0.431)	<0.001	0.279 (0.149–0.520)	0.011	0.361 (0.165–0.790)
Shock	<0.001	0.242 (0.128–0.456)	<0.001	0.255 (0.130–0.502)	0.032	0.425 (0.194–0.930)
Coma	<0.001	0.266 (0.122–0.578)	0.002	0.260 (0.112–0.605)	0.073	0.440 (0.179–1.080)
Hyperparasitemia	<0.001	0.374 (0.231–0.605)	0.001	0.427 (0.254–0.717)	0.581	0.821 (0.407–1.655)
Laboratory findings						
Mixed malaria	0.146	0.663 (0.381–1.153)	0.060	0.496 (0.239–1.031)	0.309	0.606 (0.230–1.592)
Leucocytosis	0.050	0.528 (0.278–1.001)	0.215	0.648 (0.327–1.286)	0.520	0.706 (0.245–2.035)
Thrombocytopenia < 50/nL	<0.001	0.549 (0.406–0.741)	0.010	0.621 (0.431–0.893)	0.332	0.711 (0.356–1.418)
Management						
Healthcare-associated infection	<0.001	0.266 (0.142–0.499)	<0.001	0.317 (0.168–0.601)	0.142	0.572 (0.271–1.206)
Mechanical ventilation	<0.001	0.262 (0.132–0.521)	<0.001	0.310 (0.154–0.621)	0.101	1.955 (0.878–4.353)
AKI3 (Need for RRT)	<0.001	0.125 (0.050–0.317)	<0.001	0.140 (0.055–0.357)	0.002	0.207 (0.077–0.557)
Artemisinin-based regimen	<0.001	1.593 (1.218–2.084)	0.512	1.124 (0.793–1.595)	0.837	0.932 (0.476–1.825)

Abbreviations: AKI3, acute kidney injury stage 3; APO, acute pulmonary oedema; ARDS, acute respiratory distress syndrome; CA-CCI, age-adjusted Charleson co-morbidity index; RRT, renal replacement therapy.

**Table 4 microorganisms-09-01941-t004:** Factors associated with hospital length of stay in multivariate analysis.

Covariate	Adjusted Hazard Ratio	95% Confidence Intervall	*p* Value
APO or ARDS	0.450	0.223–0.874	0.018
Shock	0.438	0.220–0.873	0.019
AKI3	0.170	0.063–0.461	<0.001

Abbreviations: AKI3, acute kidney injury grade 3 requiring renal replacement therapy; APO, acute pulmonary oedema; ARDS, acute respiratory distress syndrome.

## Data Availability

The data presented in this study are available on reasonable request only from the corresponding author. The data are not publicly available due to potential violation of the privacy of enrolled patients.

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
