# Peer review of "Factors Associated with Prolonged Hospital Length of Stay in Adults with Imported Falciparum Malaria—An Observational Study from a Tertiary Care University Hospital in Berlin, Germany"

_microorganisms, 2021, doi:10.3390/microorganisms9091941_

Round 1
Reviewer 1 Report
This is interesting data from a context (first world well-resourced hospital) that is unlike most of the apparent malaria literature. However, I think the paper needs considerable further work prior to publication.
Introduction. First paragraph appears to be about malaria generally/globally– but the first sentence of para two says ‘In Europe the situation is different…’. If the first para is not global data, where is that data from (reference 6 and 7 apparently – but will help reader to state in which countries was that work done.
What is unclear is what you mean by resource allocation. This is important it we are to see that the purpose of the study/paper has been met. Do you mean allocation of hospital rooms, ICU v other, staff and funding? If so - it is unclear how the results, model and conclusions of this work will improve future decision. How is this to be used?
While a model has been created – it has not been included in the paper. Without the model to use – how are hospitals meant to improve their decision making? E.g can the model be used to calculate from admission health parameters such as those in table 2 the likely LOS, ICU needs, staff and other resources required? If so – say so.
It remains unclear to me what is meat in the methodology by ‘repeated episodes were excluded’. Repeat episodes from what perspective – the patient’s or the hospital’s?
Do you mean only patients experiencing their first lifetime episode of malaria were included (all patients who had experienced earlier episodes of malaria were excluded)
Or do you meant that only patients experiencing their first episode at this hospital were included (all patients previously treated for malaria at the hospital were excluded).
In the first version a patient who had experienced malaria before but not been treated at this hospital before would be excluded, in the second they would be included.
Clarify what is meant by the term ‘baseline’, line 140. Normally it would be the normal ‘base’ before disease onset. E.g line 157 the patient had a pre-malaria renal disease with a baseline of >10 mg/l, that were exacerbated by subsequent malaria and presumably their levels on admissions were higher again. I think what you really mean in in line 140 and what is in table 2 are values for ‘heath status on admission ‘. In line 141-142 the is reference to ‘patents had been treated – do you mean before or after admission? I think you mean ‘patients were treated’.
Lines 145-150. 13 paitents had APO. You state APO was an indication for mechanical ventilation – yet only 9 were ventilated. Why were 4 patients not ventilated if Apo is an indication for ventilation? Is only APO beyond a certain level of severity and indication for ventilation or was indicated support not provided?
Line 167 describe more clearly the overrepresentation of LOS for the few patients with severe disease e.g. ‘ … the 55 (10.2%) patients with severe cases accounted for > 25% of total LOS…’
Where the IQR is provided as an indication of spread/variability, please consider also providing the complete range.
With regard Table 2.
Section heading demographics. Use of chemoprophylaxis, delay in presentation, and arguably history of previous malaria are not ‘demographics’- they are relevant medical history. Change heading to ‘demographics and history’.
Unclear what you mean by ‘delay in presentation’ – was someone to blame in causing a delay? Perhaps you mean ‘clinical duration prior to presentation’. In which case – have you measured the time to presentation to any doctor that may subsequently have referred them to the hospital or is this really the time of ‘clinical duration prior to admission’ (to the hospital).
Row relating to chemoprophylaxis refers only to 441/523 having regular v irregular/no use. What is the status of the other 82. If it is unknown – the values for all three states (regular v irregular/no use v unknown) should be shown.
Similarly for the following:
- ‘Destination’ the relevant state of all 535 patients should be stated e.g. Africa v SEAsia v other v unknown
- ‘Pregnancy’ the relevant state of all 535 patients should be stated.
this could be done as either ‘pregnancy status = pregnant v non-pregnant (incudes in males and non-pregnant females) or perhaps as ‘reproductive status’ = male v pregnant female v non-pregnant female. If the status of some patients is unknown – that two should indicated - ‘Relative Body weight’ the relevant state of all 535 patients should be stated. E.g obese v not obese or underweight v normal v overweight v obese v morbidly obsess etc Please clarify what is mean by obese (e.g. is it all people over normal body weight – or just those that are obese v just overweight).
- ‘BMI’ the relevant state of all 535 patients should be stated. The median and IQR is shown for 248/535. Where the others unknown?
- ‘All the laboratory finding characteristics - the relevant state of all 535 patients for each type of lab finding should be stated. E.g. the number and % with leucocytosis of 524 is indicated. Is the leucocyte status of the others unknown/test not done?
One the table incudes data for all the possible categories for each characteristic the row indicating that there were 535 in the whole group, 480 with uncomplicated malaria and 55 with severe will be redundant and can be removed – simplifying the table.
Please clarify what is meant in Table 2 by ‘hospital’ acquired infection’ – what infections – malaria? Other? What? (perhaps as a part of the table legend)
Please clarify what is meant in Table 2 by ‘mixed malaria’
Re ‘Need for medical mechanical ventilation’. Please clarify. Need or indication is one thing. Whether ventilation was actually given is another. Suspect you mean patient was ‘mechanically ventilated’.
Clarify what the p value in table 2 relates to – significance of what compared to what?
Update Table 3 to be consistent with changes in Table 2.
Figures 3 (B-D) and 4 (A-C) while O and 1 are conventions for analysis – it would be helpful it the axis labels were clearer e.g. absent v present, RRT not required v RRT required.
The discussion should relate directly to how the results deliver on the aim of this study in the broader context of relevant knowledge. There is no reference to the results of study or there meaning or value in the first 5 of 6 paragraphs of the discussion. These 5 paragraphs are instead a review of the literature, the main points of which are included in the introduction. Needs rewriting.
What of your results is meaningfully similar or different to what is in the literature?
How may your model or results fulfil the aim of the study in a way that adds to the literature in this field (if doesn’t – wrong aim). How do your model these results help hospitals improve decision making about resources? Some of the discussion seems to include recommendations about how to investigate and treat malaria. But is that the purpose aim?
Similarly for the conclusions. They need to more clearly relate to the aim of the study. This version includes some points that may belong in the rewritten discussion.
The results of the study do illustrate that care in a German well hospital has very good outcomes. Was that the aim? It also shows that people with complications stay in hospital longer – but that is hardly news.
Overall - Reconsider what does this study reveals that is new, helpful. Allocating resources, investigating potential complications and optimal clinical management are not the same things. Reconsider what are your real aim/s and what are the significant findings, what do they mean and how can that be used by others at home or elsewhere… and reframe the paper around that.
Minor corrections:
Line 11. Delete ‘also’ (also what?)
Line 19. Insert ‘decreased’ …’and decreased by 40% with each additional …..’
Line 92. Change to ‘As artesunate had was not manufactured in accordance ….. ‘
Line 111. Change to ‘(Figure 2b)’
Line 139. Change to ‘(Figure 2a)’
Author Response
Please find a detailed reply attached below (word file).

Reviewer 2 Report
This is a detailed look at the variables associated with increased length of hospital stay in imported P. falciparum cases in a tertiary hospital in Berlin, Germany. The cohort is relatively large and well described; data are robust given the presence of electronic medical records. The statistical analysis is appropriate and in-depth. Although the findings are not surprising, they are none-the-less interesting and important. Perhaps more emphasis should be given (especially in the Abstract and Conclusion) as to how these findings should be translated into clinical practice – i.e. not just “favorable outcomes can be achieved but it takes a long time” but “early identification of warning signs can lead to avoidance of complications and subsequent decrease in LOS”. This is done nicely in the Discussion section – but this concept needs to be ‘distilled’ and mentioned in the Abstract and Conclusions as well.
There are two issues that I feel should be addressed prior to publication, as well as a few minor issues.
In a paper by the same group published in 2019 in Malaria Journal, the same cohort appears to have 68 cases of severe disease identified rather than the 55 severe cases in the current report. The severe malaria criteria are identical. Clarification is needed.
Comparison should be made with the findings in Phillips et al. (Ref 11) which looked at a similar cohort in London. In that publication, Table 4 covers variables that relate to LOS. Among variables in both studies, there are different findings for the influence of platelet levels and parasitemia. These should be commented on. In addition, there are some variables not measured in the current study which were in the Phillips study (albumin, temperature, neutrophil count and respiratory rate). They should also be mentioned.
MINOR;
Line 35 – The phrase ‘so-called’ is not needed here
Line 92-94 - This sentence is awkward.
Figure 1 Legend – It should me Patients not Patient
Figure 2(a) – x-axis needs a label
Figure 2(b) – x-axis units need to be labeled as days
Table 2 – In the first line under “Co-morbidities” the “Number of chronic co-morbidities” is unmeaningful as presented. The reader is asked to compare 0 to 0 to 0 and believe that the p value is 0.005. Could this be presented in a more meaningful way? Maybe total number of chronic co-morbidities in each group.?
Table 2 – Under the “Severe malaria” section – why is hemoglobinuria included?
Table 3 – There appears to be a typographical error in the OR for Day 3 Number of Chronic 0.737 (??? - 0.912). The confidence interval does not make sense as reported.
Figure 3(D) and Figure 4(C) – should be labelled AKI3 not “RRT required”. There is currently a mismatch between the legend and the actual figure.
Line 233 – This is a confused sentence.
There appears to be an issue with the formatting of some of the references, where the second author name is dropped or truncated. This seems to occur with Refs # 6,7,9,10,15,16,17,18,21,23,24,25,31,32,35,36, and 39.

Author Response

(The authors gave the same response as above.)

Round 2
Reviewer 1 Report
I thank the author for their genuine efforts to respond to the feedback provided and the colour coding of the edits was helpful. It is an interesting report of the outcome of optimal treatment for malaria.